# MxPool: Multiplex Pooling for Hierarchical Graph Representation Learning

## Abstract

Graphs are known to have complicated structures and have myriad applications. How to utilize deep learning methods for graph classification tasks has attracted considerable research attention in the past few years. Two properties of graph data have imposed significant challenges on existing graph learning techniques. (1) Diversity: each graph has a variable size of unordered nodes and diverse node/edge types. (2) Complexity: graphs have not only node/edge features but also complex topological features. These two properties motivate us to use a multiplex structure to learn graph features in a diverse way. In this paper, we propose a simple but effective approach, MxPool, which concurrently uses multiple graph convolution networks and graph pooling networks to build hierarchical learning structure for graph representation learning tasks. Our experiments on numerous graph classification benchmarks show that our MxPool has marked superiority over other state-of-the-art graph representation learning methods. For example, MxPool achieves 92.1% accuracy on the D&D dataset while the second best method DiffPool only achieves 80.64% accuracy.

## 1 Introduction

Graphs are known to have complicated structures and have myriad of real world applications. Recently, great efforts have been put on utilizing deep learning methods for graph data analysis. Many newly proposed graph learning approaches are inspired by Convolutional Neural Networks (CNNs) (LeCun & Bengio, 1998), which have been greatly successful in learning two-dimensional image data (grid structure). The convolution and pooling layers in CNNs have been redefined to process graph data. Multitude of different Graph Convolutional Networks (GCNs) (Shuman et al., 2013) have been proposed, which can learn node level representations by aggregating feature information from neighbors (spatial-based approaches) (Hamilton et al., 2017) or by introducing filters from the perspective of graph signal processing (spectral-based approaches) (Bengio & LeCun, 2014). On the other hand, similar to the original pooling layer which comes with CNNs, graph pooling module (Defferrard et al., 2016; Zhang et al., 2018) could easily reduce the variance and computation complexity by down-sampling from original feature data, which is of vital importance, particularly for graph level classification tasks. Recently, hierarchical pooling methods that can learn hierarchical representations of graphs have been proposed (Ying et al., 2018; Gao & Ji, 2019; Lee et al., 2019) and shows state-of-the-art performance for graph classification tasks.

However, two properties of graph data have imposed significant challenges on existing graph learning techniques. 1) Diversity: each graph has a variable size of unordered nodes and has diverse node/edge types. 2) Complexity: graphs have not only node/edge features but also complex topological features. These two properties can bring troubles in both of the graph convolution operation and the graph pooling operation.

For example, when performing node-representation learning tasks (by graph convolution operation), it is enough to use small output embedding size for simple and small graphs, as shown in Figure 1(a), since large embedding size could result in overfitting problem. By contrast, it is necessary to set large output embedding sizes for complex and large graphs to learn complex graph structure properties, as shown Figure in 1(b). This creates a contradiction when processing a set of irregular graphs. For another example, when coarsening graphs (by graph pooling operation), if more attention is put on the graph structure, we may obtain a coarsened graph as shown in Figure 1(c). If more

attention is put on the node features, we may obtain another coarsened graph as shown in Figure 1(d). This creates another contradiction when processing graphs with not only node features but also topological features.

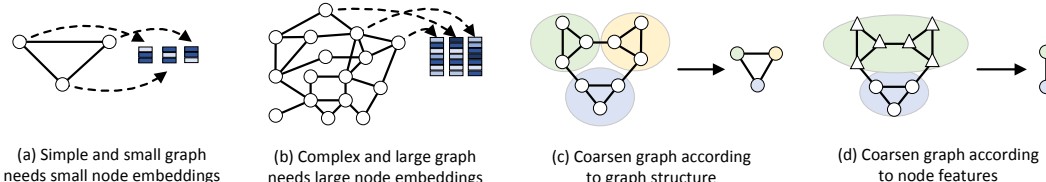

(a) Simple and small graph needs small node embeddings    (b) Complex and large graph needs large node embeddings    (c) Coarsen graph according to graph structure    (d) Coarsen graph according to node features

Figure 1: Multiple graph convolution networks with different output node embedding sizes are needed since graph sizes and complexities are not consistent, e.g., (a) and (b). Multiple graph pooling networks are needed to coarsen graphs according to different graph properties. For example, the original graph can be coarsened according to graph structure as shown in (c) or according to node features as shown in (d), where circle vertices have similar node features and triangle vertices have similar node features.

The diversity property and the complexity property of graph data motivate us to use multiplex GNN structure to learn graph features in a diverse way. On the other hand, as known, a common solution for augmenting the traditional CNN convolution layers is to use multiple convolution kernels in order to learn multiple local features. The success of CNNs on image data also inspires us to concurrently use multiple graph convolution networks and multiple graph pooling networks to learn graph representations.

In this paper, we propose MxPool in hierarchical graph representation learning for graph classification tasks[1]. MxPool comprises multiple graph convolution networks to learn node-level representations and also comprises multiple graph pooling networks to coarsen the graph. The node-level representations resulted from multiple convolution networks and the coarsened graphs resulted from multiple pooling networks are merged in a learnable way, respectively. The merged node representations and the merged coarsened graph are then used to generate a new coarsened graph, which is used in the next layer. This multiplex structure can adapt to graphs with different sizes and can extract useful information from different perspectives.

We conduct extensive experiments on numerous graph classification benchmarks and show that our MxPool has marked superiority over other state-of-the-art graph representation learning methods. For example, MxPool achieves 92.1% accuracy on the D&D dataset while the second best method DiffPool only achieves 80.64% accuracy.

## 2 RELATED WORK

In this section, we review the recent literature on GNNs, graph convolution variants, and graph pooling variants.

**Graph Neural Networks Inspired by Traditional Deep Learning Techniques.** GNNs have recently drawn considerable attention due to their superiority in a wide variety of graph related tasks, including node classification (Kipf & Welling., 2017), link prediction (Schlichtkrull et al., 2018), and graph classification (Dai et al., 2016). Many of these GNN models are inspired by traditional learning techniques. Inspired by the huge success of convolutional networks in the computer vision domain, a large number of Graph Convolutional Networks (GCNs) have emerged. Besides convolution operation, pooling operation, as another key component in CNNs, has also inspired research communities to propose graph pooling operations. There are also GNN optimizations originating from other learning approaches. Inspired by Recurrent Neural Networks (RNNs), You et al. (2018a) apply Graph RNN to the graph generation problem. DGNN (Ma et al., 2018) proposes using LSTM (Hochreiter & Schmidhuber, 1997) to learn node representations in dynamic graphs. Inspired by the attention mechanism (Vaswani et al., 2017) Graph Attention Networks (GATs) (Velickovic et al.,

---

[1]Our code is available at https://github.com/JucatL/MxPool/.

2017) introduce attentions into GCNs by differentiating the influence of neighbors. Graph AutoEncoders (GAEs) (Wang et al., 2016) origin from the autoencoder mechanism widely used for unsupervised learning and are suitable to learn node representations for graphs. GCPN (You et al., 2018b) utilizes Reinforcement Learning (RL) for goal-directed molecular graph generation.

**Graph Convolution.** Graph convolution operations fall into two categories, spectral-based approaches and spatial-based approaches. Bengio & LeCun (2014) first introduce convolution for graph data from spectral domain using the graph Laplacian matrix $L$. Besides, there exist numerous spectral-based graph convolution methods, such as ChebNet (Defferrard et al., 2016), 1stChebNet (Kipf & Welling, 2017), and AGCN (Li et al., 2018). In contrast, spatial-based convolution methods define graph convolution based on a node's spatial relations. It takes the aggregation of a node representation and its neighbors' representations to obtain a new representation for this node. In order to explore the depth and breadth of a node's receptive field, multiple graph convolution layer are stacked together, so that the features of two or more hops away neighbors can be learned. For example, GGNNs (Li et al., 2015), MPNN (Gilmer et al., 2017), GraphSage (Hamilton et al., 2017), PATCHY-SAN (Niepert et al., 2016), and DCNN (Atwood & Towsley, 2016) all fall into the spatial-based category.

**Graph Pooling.** Graph pooling operation is of vital importance for graph classification tasks (Zhang et al., 2018). It coarsens a graph into sub-graphs (Defferrard et al., 2016; Ying et al., 2018) or to sum/average over the node representations (Duvenaud et al., 2015; Gilmer et al., 2017), which can obtain a compact representation on graph level. The graph coarsening approaches obtain hierarchical graph representations either by using *deterministic* pooling methods or by using *learned* pooling methods. The deterministic pooling methods (Defferrard et al., 2016; Simonovsky & Komodakis, 2017) utilizes graph clustering algorithms to obtain next level coarsened graph that is going to be processed by GNNs, following a two-stage approach. On the other hand, the learned pooling methods (Ying et al., 2018; Lee et al., 2019; Diehl, 2019; Gao & Ji, 2019) seek to learn the hierarchical structure, which have shown to outperform deterministic pooling methods. DiffPool (Ying et al., 2018) was the first to propose learned graph pooling. It learns a soft cluster assignment matrix in layer $l$ which contains the probability values of nodes being assigned to clusters. A cluster in layer $l$ will be reduced to a node in layer $l + 1$. A GNN with input node features and adjacency matrix is used to generate the soft assignment matrix, based on which we can learn the cluster embeddings (i.e., node features in the next layer) and the coarsened adjacency matrix denoting the connectivity strength between each pair of the clusters. Besides DiffPool, numerous graph pooling methods have emerged recently, including gPool (Gao & Ji, 2019), SAGPool (Lee et al., 2019), EigenPooling (Ma et al., 2019), and Relational Pooling (Murphy et al., 2019). This paper will focus on the learned pooling.

## 3 PROPOSED METHOD

In this section, we propose MxPool to learn graph representations such that graph level classification can be applied. Before going to the details, we first introduce some notations and the problem setting.

**Problem Setting** A graph can be represented as $\mathcal{G} = \{A, F\}$, where $A \in \mathbb{R}^{n \times n}$ denotes the adjacency matrix ($n$ is the number of nodes contained in $\mathcal{G}$), and $F \in \mathbb{R}^{N \times d}$ denotes the node feature matrix ($d$ is the dimension of features). In the graph classification setting, given a set of graphs and each being associated with a label, we aim to train a model that takes an unseen graph as input and predicts its corresponding label. To make the prediction, it is important to extract useful information from multiple perspectives including both graph structure and node features.

### 3.1 OVERVIEW

MxPool is a multi-layer hierarchical GNN model. At each layer, MxPool consists of convolution operation and pooling operation. The convolution operation aims to learn node-level representations, while the pooling operation aims to learn a coarsened graph. The new coarsened graph can then be used as input to next layer. This process can be repeated several times, generating a multi-layer GNN model to learn hierarchical graph representations. The convolution operation and pooling operation are both important for graph representation learning. To simplify the illustration, we choose GCN (Kipf & Welling., 2017) as the convolution layer and DiffPool (Ying et al., 2018) as the pooling

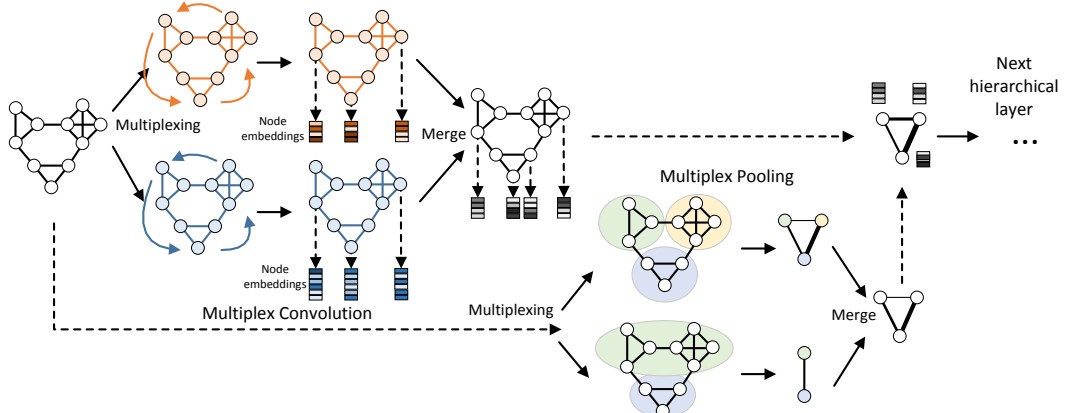

Figure 2: An illustrative example of MxPool. At each hierarchical layer, we first run multiple GCNs with different hyperparameters on the original graph, so we will have multiple diverse sets of node embeddings, which form the multiplex convolution step. Next, we use a learnable approach to merge these embeddings for each node into one. We also run multiple differentiable pooling operations on the original graph and cluster nodes together in diverse ways, which forms the multiplex pooling step. These coarsened graphs are then merged into one in a learnable approach. The merged node embeddings and the merged coarsened graph are used to generate a new coarsened graph with a new set of node features, which is going to be processed in the next hierarchical layer.

layer (which is a differentiable pooling method), but it can be extended to use other convolution variant and pooling variant as well.

Different from other hierarchical GNNs, MxPool launches multiple GCNs to learn node-level representations and also launches multiple pooling networks to coarsen the graph. The node-level representations resulted from multiple GCNs are then merged in a learnable way, and the coarsened graphs resulted from multiple pooling networks are also merged. The merged node embeddings and the merged coarsened graph are used to generate a new coarsened graph with a new set of node features. This multiplex structure can help extract useful information from different perspectives (e.g., graph structure perspective and node feature perspective) and can adapt to graphs with different sizes. We provide an illustrative example as shown in Figure 3.

The procedure of the GCN (Kipf & Welling., 2017) is to "horizontally" learn node representations, as it can only "pass message" between nodes through edges. The procedure of differentiable pooling (Ying et al., 2018) is to "vertically" summarize the node features into the higher level graph representation. The procedure of multiplexing is to "diversely" learn node representations or graph representations from different perspectives. The procedure of merging is to "synthetically" learn the diverse results and put more attention to one or more perspectives. Since the convolution, pooling, and merging operations are all differentiable, we can define an end-to-end differentiable graph representation learning framework in a hierarchical manner.

### 3.2 MULTIPLEX CONVOLUTION

In our model, we use GCN for the convolution operation. The original GCN (Kipf & Welling., 2017) is stacked by several convolutional layers, and a single convolutional layer can be written as

$$H^{(k+1)} = ReLU(\tilde{D}^{-\frac{1}{2}}\tilde{A}\tilde{D}^{-\frac{1}{2}}H^{(k)}W^{(k)}), \tag{1}$$

where $H^{(k)} \in \mathbb{R}^{n \times d}$ are the node embeddings computed after $k$ steps, $\tilde{A} = A + I$, $\tilde{D} = \sum_j \tilde{A}_{ij}$, and $W^{(k)} \in \mathbb{R}^{d \times d}$ is a trainable weighted matrix[2]. Equation (1) can be understood as a message passing process. The node embeddings $H^{(k)}$ are the "messages" transferred along edges, which are going to be used to generate new node embeddings in next round. A total number of $K$ convolutional layers

---

[2]The number of dimensions of $W$ can be different from $d$, i.e., $W \in \mathbb{R}^{d \times o}$, where $o$ denotes the output embedding's size.

are stacked to learn node representations and the output matrix $Z = H^K$ can be viewed as the final node representations learned by the GCN model.

In multi-layer GNN, suppose there are totally $L$ layers. At each layer $l$, with the input node feature matrix $X^l \in \mathbb{R}^{n^l \times d^l}$ and the adjacency matrix $A^l \in \mathbb{R}^{n^l \times n^l}$ generated from previous layer, we learn an embedding matrix $Z^{(l)} \in \mathbb{R}^{n^l \times d^{l+1}}$. Here, we use $d^{l+1}$ to denote the output embeddings' dimension since it will determine the input node embeddings $X^{(l+1)}$ at next layer $l+1$ that will be introduced later. For simplicity's sake, we will use $Z^{(l)} = GCN_c(A^{(l)}, X^l)$ to denote the GCN process (containing $K$ iterations of message passing). Initially, $A^{(0)} = A \in \mathbb{R}^{n \times n}$ is the original graph's adjacency matrix, and $X^{(0)} = F \in \mathbb{R}^{n \times d}$ is the original graph's node features.

In MxPool, we use multiple GCNs to learn multiple sets of node embeddings. These GCNs can be trained with different sets of hyperparameters $\theta$, such as weight matrix $W$'s dimension. Suppose there are $n_c$ GCNs running concurrently at each layer $l$, we will have $n_c$ sets of node embeddings $\{Z_1^{(l)}, Z_2^{(l)}, \ldots, Z_{n_c}^{(l)}\}$. Let $\theta_i$ be the hyperparameters set of the $i$th GCN. Then at layer $l$, we have node embeddings $Z_i^{(l)}$ resulted from the $i$th GCN as follows:

$$Z_i^{(l)} = GCN_c(A^{(l)}, X^l, \theta_i). \tag{2}$$

Then, the multiple sets of node embeddings $\{Z_1^{(l)}, Z_2^{(l)}, \ldots, Z_{n_c}^{(l)}\}$ are merged into one set of node embeddings $Z^{(l)}$ using a neural network:

$$Z^{(l)} = f_c(Z_1^{(l)} || Z_2^{(l)} || \ldots || Z_{n_c}^{(l)}), \tag{3}$$

where "$||$" denotes row-wise concatenation operation and $f_c()$ is a trainable neural network. One important hyperparameter of $f_c()$ is the output embeddings' dimension $d^{l+1}$, i.e., $Z^{(l)} \in \mathbb{R}^{n^l \times d^{l+1}}$. We set $d^{l+1}$ by averaging the dimensions of the multiple weight matrices $\{W_1, W_2, \ldots, W_{n_c}\}$. Suppose $W_i \in \mathbb{R}^{n^l \times d_i^l}$, we can set $d^{l+1} = \sum_{i=1}^{n_c} d_i^l / n_c$.

### 3.3 MULTIPLEX POOLING

We follow DiffPool (Ying et al., 2018) to construct our multiplex pooling layer. We learn to assign nodes to clusters at each layer $l$ using node embeddings and adjacency matrix generated from previous layer. Specifically, at each layer $l$, we learn $n_p$ cluster assignment matrices $\{S_1, S_2, \ldots, S_{n_p}\}$, and each cluster assignment matrix $S_i$ is generated as follows:

$$S_i^{(l)} = softmax\Big(GCN_p(A^{(l)}, X^{(l)}, \mu_i)\Big). \tag{4}$$

It is noticeable that $GCN_p$ is a GCN different from the $GCN_c$ used in the convolution layer, though these two GNNs consume the same input data. Each row of $S_i^{(l)}$ corresponds to one of the $n^l$ nodes at layer $l$, and each column of $S_i^{(l)}$ corresponds to one of the $c_i^l$ clusters, so that we have $S_i^{(l)} \in \mathbb{R}^{n^l \times c_i^l}$. $0 < \mu_i < 1$ denotes the hyperparameters set of the $i$th GCN. One important hyperparameter could be the compression ratio that determines the number of clusters to be assigned, i.e., $c_i^l$. Different pooling networks can use different number of clusters.

Similar to the merging process in the convolution operation, these generated assignment matrices $\{S_1^{(l)}, S_2^{(l)}, \ldots, S_{n_p}^{(l)}\}$ are merged into a single assignment matrix $S^{(l)}$ using a neural network:

$$S^{(l)} = f_p(S_1^{(l)} || S_2^{(l)} || \ldots || S_{n_p}^{(l)}), \tag{5}$$

where "$||$" denotes row-wise concatenation operation and $f_g()$ is a trainable neural network. Given the number of nodes at the next layer $l+1$, $n^{l+1}$, we should configure $f_g()$ to output an assignment matrix with $n^{l+1}$ columns, i.e., $S^{(l)} \in \mathbb{R}^{n^l \times n^{l+1}}$.

Following the pooling approach proposed in (Ying et al., 2018), we then use the merged node embeddings $Z^{(l)}$ as shown in Equation (3) and the merged assignment matrix $S^{(l)}$ as shown in Equation (5) to generate embeddings for each of the $n^{l+1}$ clusters. We also take the adjacency matrix $A^{(l)}$

and the merged assignment matrix $S^{(l)}$ to generate a coarsened adjacency matrix denoting the edge weights between each pair of cluster:

$$X^{(l+1)} = S^{(l)^T} Z^{(l)},$$
$$A^{(l+1)} = S^{(l)^T} A^{(l)} S^{(l)}, \tag{6}$$

Here, since $S^{(l)} \in \mathbb{R}^{n^l \times n^{l+1}}$ and $Z^{(l)} \in \mathbb{R}^{n^l \times d^{l+1}}$, we have the cluster embeddings $X^{(l+1)} \in \mathbb{R}^{n^{l+1} \times d^{l+1}}$. Similarly, we have the coarsened adjacency matrix $A^{(l+1)} \in \mathbb{R}^{n^{l+1} \times n^{l+1}}$

Note that, the coarsened graph is a fully connected weighted graph, so that the coarsened adjacency matrix $A^{(l+1)}$ is a real matrix and each entry in $A^{(l+1)}$ denotes the edge weight between two clusters. The cluster embeddings $X^{(l+1)}$ and the coarsened adjacency matrix $A^{(l+1)}$ will then be used as input to the next layer, where one cluster at layer $l$ corresponds to one node at layer $l + 1$.

## 4 EXPERIMENTS

In this section, we compare MxPool with the state-of-the-art graph representation learning methods in the context of graph classification task.

**Datasets.** In our experiments, we use four graph data sets chosen from benchmarks commonly used in graph classification. These include D&D (Dobson & Doig, 2003), ENZYMES (Borgwardt et al., 2005), PROTEINS (Dobson & Doig, 2003; Borgwardt et al., 2005), NCI109 (Wale et al., 2008), and COLLAB (Yanardag & Vishwanathan, 2015). Each of these datasets include hundreds to thousands graphs. The details of these datasets are provided in Table 1.

Table 1: Statistics of data sets.

| Dataset | # of graphs | # of classes | max # of nodes | avg # of nodes | avg # of edges |
|---------|-------------|--------------|----------------|----------------|----------------|
| D&D | 1178 | 2 | 5748 | 284.32 | 715.66 |
| ENZYMES | 600 | 6 | 126 | 32.63 | 62.14 |
| PROTEINS | 1113 | 2 | 620 | 39.06 | 72.82 |
| NCI109 | 4127 | 2 | 111 | 29.68 | 32.13 |
| COLLAB | 5000 | 3 | 492 | 74.49 | 2457.78 |

**Model Configurations.** We implement MxPool by modifying DiffPool. The convolution GNN model used is the "mean" variant of GraphSAGE (Hamilton et al., 2017) architecture, which is similar to the GCN and provides various aggregation methods. The pooling GNN model used is the DiffPool model. The model configurations for convolution GNN and pooling GNN is the same as DiffPool. Besides, our MxPool comprises multiple graph convolution networks and multiple graph pooling networks with different sets of hyperparameters to learn graph features from different perspectives. Besides the hyperparameters used in each convolutional/pooling GNN, the number of graph convolution networks and the number of graph pooling networks are two hyperparameters. We concurrently run 3 graph convolution networks and also concurrently run 3 graph pooling networks. The learning rate is set as 0.001. Regarding the ENZYMES dataset, since there exist 6 classes, we use cross entropy to compute the loss.

### 4.1 BASELINES AND EXPERIMENTAL SETTINGS

We consider the following state-of-the-art methods for graph classification task as baselines:

**GraphSAGE** (Hamilton et al., 2017) is a graph convolution framework proposed for semi-supervised node classification. GraphSAGE with global mean-pooling on the learned node representations can realize graph representation learning so that it can be used for graph classification task. Other graph convolution variants are omitted as empirically GraphSAGE obtained higher performance in the task (Hamilton et al., 2017).

**SortPool** (Zhang et al., 2018) is a global pooling method which uses sorting for pooling. It is built upon the GCN layer, where the features of nodes are sorted before feeding them into traditional 1D convolutional and dense layers.

Table 2: Performance comparison on graph classification.

| Baselines | D&D | ENZYMES | PROTEINS | NCI109 | COLLAB |
|---|---|---|---|---|---|
| GraphSAGE (Hamilton et al., 2017) | 75.42 | 54.25 | 70.48 | 76.50 | 68.25 |
| SortPool (Zhang et al., 2018) | 79.37 | 57.12 | 75.54 | 70.80 | 73.76 |
| gPool (Gao & Ji, 2019) | 75.01 | 48.33 | 73.63 | 66.12 | 71.12 |
| SAGPool (Lee et al., 2019) | 76.45 | - | 71.86 | 67.86 | - |
| DiffPool (Ying et al., 2018) | 80.64 | 62.53 | 76.25 | 78.86 | **75.48** |
| MxPool (Ours) | **92.10** | **66.90** | **77.21** | **82.21** | 73.44 |

**gPool** (Gao & Ji, 2019) is a recently proposed graph pooling method. It achieves pooling operation by adaptively selecting some nodes to form a smaller graph based on their scalar projection values on a trainable projection vector.

**SAGPool** (Lee et al., 2019) is a Self-Attention Graph Pooling method for GNNs in the context of hierarchical graph pooling. The self-attention mechanism is exploited to distinguish between the nodes that should be dropped and the nodes that should be retained.

**DiffPool** (Ying et al., 2018) is the first end-to-end trainable graph pooling method that learns hierarchical representations of graphs. By setting a compression ratio parameter $r$, a graph with $n$ nodes is coarsened into a graph with $n \cdot r$ nodes at each layer. We also implement our method based on DiffPool.

**Experimental Setup** In order to remove unwanted bias towards the training data, we use 10-fold cross validation for all the baselines and our approach. Since GraphSAGE and DiffPool are the two key components in our MxPool approach (GraphSAGE as the graph convolution layer and DiffPool as the graph pooling layer), we use the base implementation and hyperparameter sweeps as in our MxPool approach. Regarding the hyperparameters of SortPool, gPool, and SAGPool, we follow the same experimental setups described in their original papers. In addition, we adopt the widely used evaluation metric, i.e., accuracy, for graph classification to evaluate the performance.

## 4.2 PERFORMANCE ON GRAPH CLASSIFICATION

The graph classification performance in ter90o8ims of accuracy is reported in Table 2. For all the baselines, we use 10-fold cross validation numbers reported by the original authors if we can obtain the numbers close to the reported ones. Regarding the gPool baseline (Gao & Ji, 2019), we cannot obtain the necessary published numbers, so we use the numbers on the ENZYMES dataset and PRO-TEINS dataset reported by the third-party[3] and the number on D&D dataset reported by Lee et al. (2019). For the other datasets, we use the numbers tested by ourselves. Regarding the SAGPool baseline (Lee et al., 2019), we meet RuntimeError when processing the ENZYMES dataset and COLLAB dataset, and we denote this case by '-'.

From the table, we observe that our MxPool approach shows remarkable performance improvement over the other state-of-the-art baselines on the first four datasets. Especially on the D&D dataset, MxPool improves the performance of the second-best approach DiffPool over 11%. However, Diff-Pool performs slightly better than MxPool on the COLLAB dataset. We tested DiffPool by ourselves and could only obtain 71.78% accuracy, which is lower than the number 75.48% reported in the o-riginal paper. This may be due to the unoptimized parameter settings, but we still use the reported number 75.48% for the DiffPool baseline. In addition, GraphSAGE and DiffPool are the two basic components in our multiplex model, working as the convolution layer and the pooling layer, re-spectively. MxPool improves upon the base convolution GNN (i.e., GraphSAGE) by an average of 9.39% and improves upon the base pooling GNN (i.e., DiffPool) by an average of 3.62%.

## 4.3 ANALYSIS OF MULTIPLEX CONVOLUTION/POOLING

Our motivation for this work is to utilize multiplex hierarchical structure to deal with the diversity and complexity challenges in graph representation learning. Multiple graph convolutional networks

---

[3]https://github.com/bknyaz/graph_nn

Table 3: Effect of multiplex convolution/pooling.

| | Variations | D&D | ENZYMES | PROTEINS | NCI109 | COLLAB |
|---|---|---|---|---|---|---|
| SCSP | $c_1 * p_1$ | 80.01 | 62.17 | 75.48 | 80.07 | 71.31 |
| | $c_2 * p_1$ | 79.33 | 59.85 | 74.47 | 80.10 | 71.33 |
| | $c_3 * p_1$ | 78.75 | 60.32 | 75.14 | 78.49 | 71.14 |
| MCSP | $[c_1|c_2|c_3] * p_1$ | 88.60 | 63.01 | 75.57 | 81.43 | 72.23 |
| SCSP | $c_1 * p_1$ | 80.01 | 60.30 | 75.48 | 80.07 | 71.31 |
| | $c_1 * p_2$ | 78.64 | 56.12 | 75.96 | 78.47 | 71.78 |
| | $c_1 * p_3$ | 78.79 | 61.22 | 74.52 | 78.95 | 71.02 |
| SCMP | $c_1 * [p_1|p_2|p_3]$ | 87.50 | 61.85 | 76.01 | 78.47 | 72.89 |
| MCMP | $[c_1|c_2|c_3] * [p_1|p_2|p_3]$ | **92.10** | **66.90** | **77.21** | **82.21** | **73.44** |

(i.e., GraphSAGE) with different sets of hyperparameters are used to learn node representations. The node embedding size, as a hyperparameter in GraphSAGE, plays an important role in determining the quality of node representation. We vary the node embedding sizes in different GraphSAGE networks. On the other hand, multiple graph pooling networks (i.e., DiffPool) with different sets of hyperparameters are used to coarsen graphs. The compression ratio, as a hyperparameter in DiffPool, plays an important role in determining the quality of graph representation. We vary the compression ratios in different DiffPool networks.

In order to verify the effectiveness of multiplex convolution and multiplex pooling, we run our MxPool with single convolution network and single pooling network (SCSP), multiple convolution networks and single pooling network (MCSP), single convolution network and multiple pooling networks (SCMP), and multiple convolution networks and multiple pooling networks (MCMP), respectively. The accuracy results for graph classification are shown in Table 3. Since the suitable node embedding sizes and compression ratios are not consistent for different datasets, we use $c_1, c_2, c_3$ to denote three different graph convolution parameters (i.e., node embedding size) and $p_1, p_2, p_3$ to denote three different graph pooling parameters (i.e., compression ratio). Note that, they are different for different datasets. We have put the detailed parameter settings on our GitHub project page.

From the table, we observe that the multiplex structure significantly improves performance over the singular structure. By fixing the pooling network with $p_1$, multiplexing three convolution networks with hyperparameters $[c_1|c_2|c_3]$ performs much better than using single convolution network with either $c_1$, $c_2$, or $c_3$. A similar trend can be observed when multiplexing pooling networks. Anyhow, the best choice is to simultaneously multiplex convolution networks and multiplex pooling networks (i.e., MCMP).

### 4.4 NUMBER OF CONVOLUTION/POOLING NETWORKS

The number of convolution/pooling networks is a hyperparamter in MxPool. In the previous experiments, we use a fixed number of convolution/pooling GNNs to show the performance. In this experiment, we vary the number of convolution/pooling networks from 1 to 6 and test the performance. Our results listed in Appendix A show that the performance can be improved a lot when the number is set to 3. But as the number is increased larger, the performance is reduced. This may be because that too many networks with a large amount of parameters result in overfitting problem.

## 5 CONCLUSION

In this paper, we proposed a simple but effective multiplex GNN architecture MxPool for hierarchical graph representation learning. MxPool comprises multiple graph convolution networks to learn node-level representations and also comprises multiple graph pooling networks to coarsen the graph. The diversity challenge and the complexity challenge of graph representation learning can be well addressed in our proposed approach. Our results show that MxPool has remarkable performance improvement over the state-of-the-art graph representation learning methods. Future work includes designing unpooling layers to form an encoder-decoder learning structure to deal with node classification tasks and link prediction tasks.

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

## A   RESULTS OF THE NUMBER OF CONVOLUTION/POOLING NETWORKS

In this experiment, we vary the number of convolution/pooling networks from 1 to 6. The number of convolution networks and the number of pooling networks are the same. The graph classification accuracy results on D&D, ENZYMES, and PROTEINS datasets are shown in Table 4.

Table 4: Effect of number of convolution/pooling GNNs.

|  | **1** | **2** | **3** | **4** | **5** | **6** |
|---|---|---|---|---|---|---|
| D&D | 80.24 | 88.57 | **92.10** | 89.29 | 87.86 | 87.32 |
| ENZYMES | 62.02 | 60.80 | **66.90** | 64.89 | 65.71 | 62.70 |
| PROTEINS | 76.25 | **77.31** | 77.21 | 76.73 | 75.58 | 75.19 |

From the table, we can see that the best performance is achieved when the number is set as 2 or 3. But as the number is increased larger than 3, the performance is reduced. This may be because that too many networks with a large amount of parameters result in overfitting problem.

