# OpenReview forum: "MxPool: Multiplex Pooling for Hierarchical Graph Representation Learning"
_ICLR.cc/2020/Conference — Reject_

### Official Review · AnonReviewer1 · 2019-10-22
**Official Blind Review #1**

**Rating:** 3

**Review:**

Strengths:
-- The paper is well written and easy to follow
-- Learning graph representation learning is a very important problem
-- The performance of the proposed approach are strong on the existing data sets

Weakness
--  the novelty of the proposed method is marginal
-- Some real-case study on why the model works are not presented.
-- The data sets used in the experiments are too small

This paper studied learning the hierarchical representation of graphs. Comparing to the previous work DiffPool, the author proposed to use multiple node representations and multiple pooling functions. Experimental results on a few data sets prove the effectiveness of the proposed approach.

Overall, the paper is well written and easy to follow. Learning the hierarchical representation of graphs is an important and also interesting problem. However, I feel the novelty of the proposed method is marginal. And besides, though illustrated using artificial example in the introduction, it is not clear why the proposed method work in practice. Specifically, I have the following questions:
(1) How does multiple convolutional method compared with just concatenating the weight matrix W_i, in other words, we can just use a much larger node embedding size. From equation (2) and (3), I feel these two are identical. The authors should explain this.
(2) Similar concerns in the multiple pooling case.
(3) It would be more convincing that the authors presented a few real cases to prove the effectiveness of the proposed approach instead of only using the artificial examples in introduction.

**Experience Assessment:**

I have published in this field for several years.

**Review Assessment: Checking Correctness Of Derivations And Theory:**

I carefully checked the derivations and theory.

**Review Assessment: Checking Correctness Of Experiments:**

I carefully checked the experiments.

**Review Assessment: Thoroughness In Paper Reading:**

I read the paper thoroughly.

---

### Official Review · AnonReviewer3 · 2019-10-22
**Official Blind Review #3**

**Rating:** 3

**Review:**

This paper extends DiffPool for hierarchical graph representation learning (in particular, graph classification). The authors empirically show that for several data sets, the approach outperforms quite a few recently proposed strong competitors.

The proposed approach is reasonable, but is not much innovative. The prior work DiffPool uses GCNs to parameterize the node embedding matrix Z and the cluster assignment matrix S. This paper computes a number of Zs and Ss, each of which result from a different hyperparameter choice of GCN, and then combines them through concatenation and feed forward transform. Significance of the contribution is a bit marginal.

The experimental results appear to be exciting, in light of the substantial boost in classification accuracy on the D&D data set. However, the experiment design and the reporting of results are doubtful. A major concern is the copying of results reported by prior work to Table 2. It is unclear whether these numbers were obtained from the same experiment setting. For example, the number for SAGPool + D&D comes from Lee et al., 2019, but the DiffPool number in that paper is 66.95, which is significantly different the one shown here, 80.64, copied from Ying et al., 2018.

Another concern is the missing numbers for SAGPool. Although the authors explain the difficulty of obtaining these numbers, a lack of them does not complete the empirical evaluation.

Minor comments/questions:

- In the first sentence of 4.2, the word "ter90o8ims" is a typo of "terms".

- What are the neural networks f_c in (3) and f_p in (5)?

- Texts after (5) read f_g instead of f_p.


**Experience Assessment:**

I have published one or two papers in this area.

**Review Assessment: Checking Correctness Of Derivations And Theory:**

I carefully checked the derivations and theory.

**Review Assessment: Checking Correctness Of Experiments:**

I carefully checked the experiments.

**Review Assessment: Thoroughness In Paper Reading:**

I read the paper at least twice and used my best judgement in assessing the paper.

---

### Official Review · AnonReviewer2 · 2019-10-23
**Official Blind Review #2**

**Rating:** 3

**Review:**

This paper introduces a new hierarchical graph representation learning method for graph classification. It builds upon the diffpool method. Specifically, the authors propose the multiplex convolution and the multiplex pooling operation. The multiplex convolution learns multiple graph convolution operations (potentially with different parameters) and merges them using a neural network. Similarly, the multiplex pooling learns multiple assignment matrices utilizing diffpool and merges them using a neural network. The effectiveness of the proposed model is demonstrated with experiments in several standard benchmark datasets for graph classification.

Strengths

The overall architecture of the proposed method can be regarded as an extension of diffpool method by using multiple convolution and pooling operations. An analysis on how the number of convolution and pooling operations can affect the performance of the model shows the effectiveness of the proposed multiplex convolution/pooling operations. The effectiveness of the proposed model is further verified by the experimental results on 5 standard benchmark datasets.

Weakness

The novelty of the paper is limited. The major contribution of this paper is to utilize multiple convolution/pooling operations and merge them with neural networks.

It is not clear how issues mentioned in the motivational examples shown in Figure 1 can be solved by the proposed model. As discussed in Figure 1(a) and Figure 1(b), small graphs may prefer small embedding while large graphs may prefer large embedding. However, it is unclear how the proposed model achieves this goal as the combination of the embeddings from different convolution operations is through a neural network that is shared by all graphs. Similarly, for Figure 1(c) and Figure 1(d), it is not clear how the proposed model can focus more on one type of information than the other. It would be helpful if there are some empirical results to demonstrate this.

Recommendations

It would be better if the authors can provide some analysis on time complexity of the proposed model. The major concern of the efficiency is the introducing of fully connected graph after each pooling as mentioned in Section 3.3.

Minor comments

In section 3.3, “is a GCN different” -> “is a GCN variant”?
In Section 3.3, should there be a softmax function after Eq.(5)?
In section 4.2, “in ter90o8ims” -> “in terms”?




**Experience Assessment:**

I have published in this field for several years.

**Review Assessment: Checking Correctness Of Derivations And Theory:**

I assessed the sensibility of the derivations and theory.

**Review Assessment: Checking Correctness Of Experiments:**

I carefully checked the experiments.

**Review Assessment: Thoroughness In Paper Reading:**

I read the paper thoroughly.

---

### Decision · Program_Chairs · 2019-12-19

**Decision:**

Reject

**Comment:**

All three reviewers are consistently negative on this paper. Thus a reject is recommended.